# One Brain—All Cells: A Comprehensive Protocol to Isolate All Principal CNS-Resident Cell Types from Brain and Spinal Cord of Adult Healthy and EAE Mice

**DOI:** 10.3390/cells10030651

**Published:** 2021-03-15

**Authors:** Christina B. Schroeter, Alexander M. Herrmann, Stefanie Bock, Anna Vogelsang, Susann Eichler, Philipp Albrecht, Sven G. Meuth, Tobias Ruck

**Affiliations:** 1Department of Neurology with Institute of Translational Neurology, University Hospital Muenster, 48149 Muenster, Germany; christina.schroeter@ukmuenster.de (C.B.S.); stefanie.bock@ukmuenster.de (S.B.); anna.vogelsang@ukmuenster.de (A.V.); susann.eichler@ukmuenster.de (S.E.); 2Department of Neurology, University of Düsseldorf, 40225 Duesseldorf, Germany; alexander.herrmann@med.uni-duesseldorf.de (A.M.H.); phil.albrecht@gmail.com (P.A.)

**Keywords:** single-cell isolation, demyelinating autoimmune diseases, astrocytes, microglia, neurons, oligodendrocytes, MACS, FACS, EAE, CNS

## Abstract

In experimental autoimmune encephalomyelitis (EAE), an animal model of multiple sclerosis, the role of each central nervous system (CNS)-resident cell type during inflammation, neurodegeneration, and remission has been frequently addressed. Although protocols for the isolation of different individual CNS-resident cell types exist, none can harvest all of them within a single experiment. In addition, isolation of individual cells is more demanding in adult mice and even more so from the inflamed CNS. Here, we present a protocol for the simultaneous purification of viable single-cell suspensions of all principal CNS-resident cell types (microglia, oligodendrocytes, astrocytes, and neurons) from adult mice—applicable in healthy mice as well as in EAE. After dissociation of the brain and spinal cord from adult mice, microglia, oligodendrocytes, astrocytes and, neurons were isolated via magnetic-activated cell sorting (MACS). Validations comprised flow cytometry, immunocytochemistry, as well as functional analyses (immunoassay and Sholl analysis). The purity of each cell isolation averaged 90%. All cells displayed cell-type-specific morphologies and expressed specific surface markers. In conclusion, this new protocol for the simultaneous isolation of all major CNS-resident cell types from one CNS offers a sophisticated and comprehensive way to investigate complex cellular networks ex vivo and simultaneously reduce mice numbers to be sacrificed.

## 1. Introduction

Multiple sclerosis (MS) is a chronic inflammatory autoimmune disease affecting the central nervous system via demyelination accompanied by axonal damage and neurodegeneration. Although extensively studied, its pathophysiology is not yet fully understood [1,2,3,4]. Mice have become the most frequently used animal models due to the broad spectrum of available techniques for the generation of transgenic or knockout mice. Myelin oligodendrocyte glycoprotein 35–55 (MOG_35–55_) induced experimental autoimmune encephalomyelitis (EAE) represents the most common animal model for MS nowadays since it shares many of its critical clinical and pathophysiological features and is especially useful to study neuroinflammatory pathways [5,6,7,8,9,10,11].

To investigate molecular and cellular mechanisms in autoimmune diseases, the generation of reproducible and sophisticated cell isolation protocols is indispensable. However, the majority of the current technologies to isolate and analyze CNS-resident cells show some serious shortcomings. Very sophisticated methods for the isolation of cells of one single type exist [12,13,14,15]; others are established for postnatal mice [16,17,18,19,20,21,22]. Also, many of the pre-existing protocols result in an impaired cell function preventing further cultivation and thereby further experimental assays. Still, others try to preserve the cellular functionality at the cost of purity by isolating and cultivating the cell type of interest in co-culture with its physiologically neighboring cells [23,24,25,26,27,28]. However, over the last years, many cutting-edge protocols have been developed for the differentiation of these cell types from neural stem cells or progenitors, which have enabled the detailed study of molecular mechanisms in vitro [14,29,30,31,32,33,34]. Unfortunately, these protocols cannot sufficiently address complex intercellular networks and ultimately can only be investigated in an in vivo system.

Hence, our aim was to create a robust and reproducible protocol for simultaneous generation of pure, viable single-cell suspensions of principal CNS-resident cell types (microglia, oligodendrocytes, astrocytes, and neurons) from adult mice—applicable in healthy mice as well as in EAE.

After dissociation of the brain and spinal cord from adult mice, microglia, oligodendrocytes, astrocytes, and neurons were isolated via magnetic-activated cell sorting MACS [35]. Here, the separation of different cell types can be achieved in two principal ways: cells are sorted out by magnetic labeling of their cell-type-specific surface markers, while unlabeled cells pass through, resulting in the positive selection of certain cell types [35]. By contrast, undesired cells may be biotinylated and depleted via magnetic labeling (negative selection).

Flow cytometry was the mainstay to evaluate the purity and viability of the resulting single-cell suspensions, commonly aiming at purity and viability above 90% and 80%, respectively. Additionally, immunofluorescence stainings assessed the cell-type-specific morphology and expression of typical in vivo markers. Responsiveness of cells to inflammatory stimuli was proven by upregulation of the cytokines IL-6 and TNF-*α* upon stimulation with LPS. Sholl morphological analyses [36] at different time points of cultivation were established to prove a physiological cell-type-specific morphological development with regards to growth and ramification. 

In conclusion, we strived to establish a protocol for the simultaneous isolation of all four major CNS-resident cell types in healthy and EAE mice that offers a helpful tool for research groups studying neuroinflammatory pathways allowing a more accurate analysis of complex biomolecular mechanisms and cellular networks ex vivo.

## 2. Materials and Methods

### 2.1. Mice

All experiments were performed with 10 to 20-week-old female C57BL/6J mice (Charles River Laboratories, Sulzfeld, Germany). Mice were kept under IVC (individually ventilated cages) animal housing conditions.

### 2.2. Active EAE Model

Experiments were approved by local authorities (Landesamt für Natur, Umwelt und Verbraucherschutz Nordrhein-Westfalen; 81-02.04.2018.A266) and carried out following the German and EU animal protection law. 

Active EAE was induced in female C57BL/6J mice at the age of 10–12 weeks by immunization with MOG_35–55_ peptide (Charité, Berlin, Germany) as previously described. Mice were anesthetized with isoflurane (AbbVie, North Chicago, IL, USA) and subcutaneously immunized with an emulsion consisting of 200 µg MOG_35–55_ and 200 µL complete Freund’s adjuvant (Merck KGaA, Darmstadt, Germany) including 200 µg Mycobacterium tuberculosis (strain H37 Ra; Becton, Dickinson and Company (BD), Franklin Lakes, NJ, USA). After 2 h, the mice were injected intraperitoneally with 100 ng Pertussis toxin (PTx; Hooke Laboratories Inc., Lawrence, MA, USA) dissolved in 100 µL 1X PBS. PTx injections were repeated on day 2 after immunization. 

The weight and clinical score of each mouse were evaluated daily by two blinded investigators according to the following scoring system: grade 0—no clinical signs of EAE, grade 1—limp tail tip, grade 2—limp tail, grade 3—moderate hindlimb weakness and uncoordinated gait, grade 4—complete hindlimb weakness and ataxic gait, grade 5—mild paraparesis of hindlimbs, grade 6—paraparesis, grade 7—paraplegia, grade 8—tetraparesis, grade 9—quadriplegia, and grade 10—death. The mean cumulative score was calculated as the sum of the daily scores of all mice until the end of the experiment divided by the number of animals. Mice with a weight loss exceeding 20% of their initial body weight or a clinical score >7 would have been taken out of the experiment. EAE mice were euthanized at disease maximum (16 days after EAE induction) for the preparation of the brain and spinal cord.

### 2.3. Isolation of Murine CNS-Resident Cells

All following reagent volumes are given for the range of 20 mg to 500 mg of neural tissue, that is, one adult murine brain and spinal cord. For the dissociation of more than one CNS, all reagent volumes and materials were upscaled accordingly. If cell culture experiments were planned subsequent to the cell isolation, all steps were performed under sterile conditions. Buffers were degassed and stored on ice. Only pre-cooled solutions were applied. Vortexing was avoided throughout the whole protocol.

#### 2.3.1. Dissection of the CNS

After termination with carbon dioxide, each mouse was perfused twice with 20 mL PBS. The spinal cord was flushed out of the spinal canal with PBS and cut into 0.5 cm long segments using a scalpel. The brain was removed carefully and cut into 8–10 sagittal slices with the help of a murine brain matrix (Ted Pella, Redding, CA, USA). The CNS tissue of one mouse was pooled in a petri dish filled with D-PBS (Dulbecco’s Phosphate Buffered Saline (1X) with calcium and magnesium, supplemented with 1 g/L glucose and 36 mg/L sodium pyruvate). The dishes were stored on ice until further downstream processing.

#### 2.3.2. CNS Tissue Dissociation

The tissue dissociation was performed with the Adult Brain Dissociation Kit, mouse, and rat (Miltenyi Biotec, Bergisch Gladbach, NRW, Germany), following the manufacturer’s instructions. The CNS tissue of each mouse was transferred to a gentleMACS C Tube (Miltenyi Biotec) containing both enzyme mixes from the kit for enzymatic digestion. Mechanical enzymatic tissue dissociation was performed using program 37C_ABDK_01 of the gentleMACS Octo Dissociator with Heaters (Miltenyi Biotec). After dissociation, each CNS homogenate was applied to one 70 µm cell strainer (Corning, Corning, MA, USA).

#### 2.3.3. Debris and Red Blood Cell Removal

Debris and red blood cell removal were performed according to the kit’s protocol (Adult Brain Dissociation Kit, Miltenyi Biotec). Contrary to the manufacturer’s instructions, the brake was turned off during centrifugation of the density gradient as this resulted in a more precise separation of the three phases (bottom—cell suspension—myelin—supernatant with the debris of a lower density than targeted cells—top) allowing the reliable removal of all myelin residues. Red blood cells were removed subsequently via an osmotic gradient applying the provided Red Blood Cell Removal Solution at a 1:10 dilution in double-distilled water (ddH_2_O). The remaining cells were washed and resuspended in 80 µL PB buffer (Dulbecco’s Phosphate Buffered Saline (1X) without calcium and magnesium, supplemented with 0.5% bovine serum albumin) per CNS homogenate. When processing more than one mouse per experimental condition, up to two CNS homogenates were pooled before debris removal (maximum weight of neural tissue was 1000 mg), and volumes were upscaled following the company’s directions. 

#### 2.3.4. MACS in Naïve and EAE Mice

The cell count was determined by the use of a Neubauer improved counting chamber (Hecht Assistant, Rhön-Grabfeld, Germany). For this purpose, a fraction of the CNS homogenate was diluted 1:50 in PB buffer, followed by a further dilution of 1:10 in 0.4% trypan blue solution (Thermo Fisher Scientific, Waltham, MA, USA). 

The purified undiluted CNS homogenate was divided into two fractions for the concurrent isolation of microglia and oligodendrocytes. The ratio of both fractions was determined depending on the desired amount of each cell type. Table 1 contains the detailed and structured protocol illustrating the simultaneous isolation steps in rows.

The negative flow-through of the oligodendrocytes was centrifugated at 300 g and 4 °C for 10 min. The supernatant was carefully removed. The cell pellet was resuspended with 80 µL PB buffer per CNS homogenate that was initially processed for the oligodendrocyte isolation. The total count of O4^−^ cells was determined with the help of a Neubauer improved counting chamber after diluting a cell fraction 1:50 in PB buffer and then 1:10 in 0.4% trypan blue solution.

The purified undiluted CNS homogenate was divided into two fractions for the simultaneous isolation of neurons and astrocytes. The ratio of both fractions was determined depending on the desired amount of each cell type. Table 2 contains the detailed and structured protocol illustrating the synchronous isolation steps in rows. 

#### 2.3.5. Amendment of the Protocol for Isolation of CNS-Resident Cells from EAE Mice

To eliminate CD11b^+^ cell populations other than microglia (mainly monocytes, macrophages, dendritic cells, natural killer cells, and granulocytes) from the CD11b^+^ cell fraction isolated from EAE mice, the MACS-based isolation protocol had to be complemented by fluorescence-activated cell sorting (FACS). For this purpose, the cells were centrifugated at 400× *g* at room temperature (RT) for 5 min. The supernatant was carefully aspirated, and the cells were resuspended with 100 µL of 1X PBS supplemented by both fluorochrome-conjugated monoclonal antibodies CD11b FITC (clone M1/70, 1:50, BD Biosciences, San Jose, CA, USA) and CD45 BV510 (clone 30-F11, 1:100, BioLegend, London, UK). The concentrations of both antibodies were carefully titrated prior to experiments. Single stainings, as well as unstained samples, were used for compensation. After incubation for 15 min in the dark at RT, the reaction was stopped by adding 500 µL PBS followed by another centrifugation step. The cell pellet was resuspended with 300 µL 1X PBS containing DNAse at a final concentration of 10 µg/mL (Merck KGaA). The cells were stored at 4 °C until sorting. Immediately before sorting, the cell suspension was applied on a 100 µm strainer placed on a new FACS tube. Subsequently, the strainer was rinsed with 1 mL 1X PBS supplemented with 10 µg/mL DNAse, and the tube was positioned in the sorter BD FACSAria III (BD Biosciences). The flow rate was set to 1,000 events per second; the 100 µm nozzle was utilized. The target cell population of CD45^int^CD11b^high^ cells was sorted into new 15 mL falcons containing pre-warmed microglia medium.

### 2.4. Purity Analyses of Isolated CNS-Resident Cells

Before and after isolation, all four CNS-resident cell types were assayed with the same flow cytometry panel to assess and compare their purity and viability. We used 2 × 10^6^ isolated cells per staining. The panel comprised the following fluorochrome-conjugated monoclonal antibodies for the detection of cell-type-specific surface markers: CD11b FITC (clone 1/70, 1:100, BD Biosciences), Biotin-PE (clone Bio3-18E7, 1:400, Miltenyi Biotec), ACSA-2 PE-Vio615 (clone REA-969, 1:200, Miltenyi Biotec), O4 APC (clone REA-576, 1:400, Miltenyi Biotec), and CD45 BV510 (clone 30-F11, 1:150, BioLegend). One microgram of anti-CD16/32 antibody (BioLegend) was added per 1 × 10^6^ cells for blocking of Fc receptors. Live/dead cell discrimination was performed with the help of eBioscience Fixable Viability Dye eFluor 780 (1:10,000, Thermo Fisher Scientific). All antibody concentrations were carefully titrated prior to experiments. Cells were stained for 15 min in the dark at RT and washed once with 500 µL PBS followed by centrifugation. 

Another fraction of each single-cell suspension was used for intracellular staining of NeuN (NeuN AF647, clone EPR12763, 1:200, Abcam, Cambridge, UK), a neuron-specific nuclear marker. For this purpose, 2 × 10^6^ cells of each single-cell suspension were fixed and permeabilized with the eBioscience Foxp3/Transcription factor staining buffer set (Thermo Fisher Scientific) following the manufacturer’s instructions.

Finally, stained cells were resuspended in 300 µL 1X PBS and analyzed by a CytoFLEX S (Beckman Coulter, Krefeld, Germany) using Kaluza software V2.1.1 (Beckman Coulter). Flow cytometry compensations were set beforehand with eBioscience OneComp eBeads and the ArC Amine Reactive Compensation Bead Kit (Thermo Fisher Scientific). Corresponding single stainings, Fluorescence Minus One (FMO) controls as well as unstained samples were used for further compensations and data interpretation.

### 2.5. Cultivation of Isolated CNS-Resident Cells

Table 3 contains the necessary preparations and handling for the cultivation of the CNS-resident cells upon isolation.

### 2.6. Validation of Isolated CNS-Resident Cells

All four CNS-resident cell types were validated morphologically as well as on a functional level.

#### 2.6.1. Immunocytochemistry

For histological validation, cultivated CNS-resident cells were stained with a cell-type specific marker. For that purpose, on the day of processing, cells were washed once with 1X PBS before they were fixed with 4% paraformaldehyde for 15 min at RT. After three washing steps, cells were blocked for 1 h at RT with a solution consisting of 5% BSA, 1% host serum, and 0.2% Triton-X (Merck KGaA) in 1X PBS in order to avoid false-positive results. After blocking, the incubation with the primary cell-type-specific antibody (Table 4) was performed overnight at 4 °C in a similar blocking solution lacking Triton-X. On the next morning, the cells were washed three times with 1X PBS before being incubated with the corresponding fluorophore-conjugated secondary antibody (Table 4) in 1% BSA for 1 h at RT in the dark. After three washing steps, the cells were mounted on coverslips with 10 µL Fluoromount-G containing DAPI (Thermo Fisher Scientific) and stored at 4 °C in the dark. On the next morning, fluorescence images were acquired with a Zeiss Axio Scope.A1 (Zeiss, Göttingen, Germany) using 40-fold objectives.

#### 2.6.2. Enzyme-Linked Immunosorbent Assays

For functional validation, cultivated microglia and astrocytes were stimulated for 48 h with 100 ng/mL lipopolysaccharide (LPS, Sigma, St. Louis, MO, USA). The supernatant was then tested for IL-6 and TNF-*α* expression levels via an IL-6 Mouse Uncoated ELISA kit and a TNF alpha Uncoated ELISA (Thermo Fisher Scientific) according to the manufacturer’s instructions. Beforehand, we performed dilution series for both cytokines and cell types. While no further dilution of the astrocytes’ supernatants was necessary, the supernatants of the microglia were diluted 1:60 for the final assays. Samples were measured in technical duplicates with the Tecan plate reader Infinite M200 Pro (Tecan, Männedorf, Schweiz). We acquired five biological replicates per experimental condition.

#### 2.6.3. Sholl Analysis

In order to demonstrate that oligodendrocytes and neurons were still viable and functional after MACS, Sholl analyses [36] were performed at different time points of their cultivation period. Here, an algorithm first created concentric shells around the cell centers. Then, the number of intersections of these shells with the cell processes was counted. In this way, cell growth and ramification could be depicted, illustrating morphological development over time. Microscopic images were acquired using a Zeiss Axio Scope.A1 with 40-fold objectives. Image analysis was performed using the Sholl analysis plugin for Fiji [37]. The radius step size was set to 0.5 µm. We analyzed five biological replicates per time point. For each biological replicate and time point, the mean of five cells was calculated.

#### 2.6.4. Ly6 Staining

Microglia were gated as CD45^int^CD11b^high^. In order to prove that the resulting cell fraction was not contaminated by other CNS-resident myeloid populations, we performed a flow cytometry staining of lymphocyte antigen 6 (Ly6). The panel comprised the following fluorochrome-conjugated monoclonal antibodies for the detection of cell-type-specific surface markers: Ly6G BV421 (clone 1A8, 1:100, BioLegend), CD11b FITC, Ly6C APC (clone HK1.4, 1:200, BioLegend), and CD45 BV510. One microgram of anti-CD16/32 antibody was added per 1 × 10^6^ cells for blocking of Fc receptors. Live/dead cell discrimination was performed with the help of eBioscience Fixable Viability Dye eFluor 780. All antibody concentrations were carefully titrated prior to experiments. Cells were stained for 15 min in the dark at RT and washed once with 500 µL PBS followed by centrifugation. 

Finally, stained cells were resuspended in 300 µL 1X PBS and analyzed by a CytoFLEX S using Kaluza software. Flow cytometry compensations were set beforehand with eBioscience OneComp eBeads and the ArC Amine Reactive Compensation Bead Kit. Corresponding single stainings, FMO controls as well as unstained samples were used for further compensations and data interpretation. Neutrophils (gated as CD45^high^CD11b^+^) served as positive control.

### 2.7. Statistical Analysis

For each type of experiment, group sizes are given in the figure legends. Statistical analyses and graphs were prepared using Prism 8.4.3 (Graph Pad, San Diego, CA, USA). Data were presented as the mean ± SEM. The significance level was set to a *p*-value < 0.05. Statistical analysis was performed using the non-parametric Mann–Whitney U test.

## 3. Results

### 3.1. All Principal CNS-Resident Cell Types can Be Isolated Simultaneously from One CNS Homogenate with a Purity of Around 90%

In order to reduce the number of mice needed for experiments and to increase the comparability of molecular analyses on a cellular level, our goal was to simultaneously isolate all four principal CNS-resident cell types from the same CNS homogenate. For this purpose, we modified pre-existing MACS protocols (Miltenyi Biotec), extended and rearranged them so that all cells could be isolated simultaneously. To date, it was only possible to isolate one of these cell types at once using separate CNS homogenates for each cell type. With the newly established protocol, we were able to simultaneously isolate microglia, oligodendrocytes, astrocytes, and neurons via MACS from the same CNS homogenate (Figure 1). We used anti-CD11b MicroBeads for the isolation of microglia, anti-O4 MicroBeads for oligodendrocytes, and anti-ACSA-2 MicroBeads for isolating astrocytes (positive selection). For neurons, all non-neuronal cells were biotinylated, magnetically labeled, and depleted via the same mechanism (negative selection). 

For reliable purity analysis of the generated single-cell suspensions, we established two new flow cytometry purity panels consisting of surface markers specific for each cell type along with live/dead cell discrimination (Figure 1). We used individual FMO gating for all cell types. FMO controls were obtained from the CNS homogenate before MACS still containing all four CNS-resident cell types (Figure 2a). However, as Figure 2 illustrates, the fluorescence intensity signals of the cell-type-specific markers and FMO controls showed some overlap between the four isolated CNS-resident cell-types; therefore, we modified the individual gating strategy to prevent overlap (Figure 3, Figure 4 and Figure 5 and Appendix A). 

Initially, we applied both purity panels to the purified CNS homogenate before magnetic separation resulting in a phenotypic characterization pre-isolation illustrating its heterogeneity (Figure 2a). Post-isolation, resulting cell yields per mouse and cell-type were investigated: Upon simultaneous isolation from one naïve mouse, an average of 840,000 microglia, 3,230,000 oligodendrocytes, alongside 860,000 astrocytes and 450,000 neurons could be isolated (Figure 2b). In EAE mice, due to additional sorting subsequent to MACS, the microglia cell yield per mouse dropped to a mean of 170,000 microglia after eliminating all other CD11b^+^ cells (Figure 2b). Apart from that, approximately 2,900,000 oligodendrocytes, 520,000 astrocytes, and 440,000 neurons could be isolated when dissociating the CNS of one EAE mouse. 

After isolation, we initially performed FSC-A/SSC-A gatings for each CNS-resident cell type to illustrate its granularity and cell size, which both varied a lot between the four isolated cell populations. The FSC-A/SSC-A gating was not used for the remaining flow cytometry analyses. Instead, all further purity analyses were obtained from the respective live cell gates. Phenotypic characterization post-isolation showed that we were able to acquire viable single-cell suspensions of all four major CNS-resident cell types with a purity of around 90%. Microglia were isolated via magnetic MicroBeads binding to the surface marker CD11b. Flow cytometry analysis confirmed cellular viability at 85.65% and purity at 93.31%. Microglia were gated as being CD45^int^ (93.31%) and CD11b^high^ (96.11%) according to the literature [38,39,40] (Figure 3a). Simultaneously to microglia, oligodendrocytes were segregated from the remaining CNS homogenate by anti-O4 MicroBeads. Within the generated single-cell suspension, 78.01% of cells were viable, and 91.41% expressed the oligodendrocyte-specific surface marker O4 (Figure 3b). 

Flow cytometry purity analysis of astrocytes isolated straight from the initial CNS homogenate with the help of anti-ACSA-2 MicroBeads showed 33.31% had relevant contamination with oligodendrocytes (Figure 4a). The same was true for neurons isolated by biotinylation and consecutive depletion of all non-neuronal cells (33.15% oligodendrocytes) (Figure 5a). Here, we used NeuN as a cell-type specific nuclear marker. Since a vast majority of the contaminating cell populations was O4^+^, we assumed that either living or dead oligodendrocytes stuck tightly to astrocytes and neurons and were therefore isolated along with these two cell fractions. Isolating neurons and astrocytes simultaneously from the negative flow-through of oligodendrocytes led to a sufficient depletion of O4^+^ cells: While the purity of astrocytes increased by over 30% from 59.87% to 89.23% (Figure 4), the purity of neurons could be improved from 66.77% to 81.25% (Figure 5). 

In order to check for negative side effects of the sequential isolation on the viability of the cells, we compared the percentages of viable cells after immediate MACS isolation to those after depleting oligodendrocytes first: For astrocytes, the viability even mildly increased from 79.36% living cells without O4^+^ depletion to 80.56% viable astrocytes isolated from the O4^−^ flow-through (Figure 4). The same was true neurons (78.24% without versus 83.90% with prior depletion of oligodendrocytes) (Figure 5).

### 3.2. Isolated CNS-Resident Cells Showed a Cell-Type Specific Morphology and Functionality during Cultivation

In order to validate the isolated CNS-resident cell types on histological and functional levels, we combined immunocytochemistry with further functional analyses. For immunofluorescence staining of microglia, we used primary antibodies directed against Iba-1 and CD11b [38,44,45] (Figure 6a, Appendix A). As an oligodendroglial marker, we used NOGO-A [46] (Figure 6c). Astrocytes were immunofluorescently labeled, showing their expression of GFAP [47] (Figure 6e). MAP2 served as a cell-type-specific cytoskeletal marker for neurons [48] (Figure 6g). All four isolated CNS-resident cell types showed a cell-type-specific morphology and cellular marker expression. 

For evaluation of the cellular viability on a functional level, we performed IL-6 and TNF-*α* ELISAs for microglia and astrocytes. After stimulation with 100 ng/mL LPS for 48 h, the expression levels of the pro-inflammatory cytokines IL-6 and TNF-*α* were significantly increased in both cell types (Mann–Whitney U test, *p*-value 0.0079) (Figure 6b,f). As a functional readout for oligodendrocytes and neurons, Sholl analyses were carried out during cultivation. Here, not only physiological cell growth but also a typical ramification could be observed over time, demonstrating that the cells were still viable after the MACS procedure and could be cultivated for further experiments (Figure 6d,h).

### 3.3. In EAE Mice, Immigration of Peripheral Immune Cells into the CNS Necessitates Sorting of Microglia from the CD11b^+^ Cell Population after MACS

In order to study the transferability of the newly established protocol on disease models of neuroinflammation, we applied it to an animal model of EAE (Figure 7). Mice were sacrificed at disease maximum (here: day 16) (Figure 7a). In the flow cytometry purity analysis of isolated microglia, it became apparent that in a setting of neuroinflammation, other CD11b^+^ immune cells like monocytes, neutrophils, natural killer cells, granulocytes, and macrophages were isolated along with the microglia. It is well-known that these cells immigrate into the CNS during the EAE course [41,42,43]. Hence, microglia had to be sorted as CD45^int^CD11b^high^ cells from the CD11b^+^ single-cell suspension after MACS. In this way, purity could be increased from 71.60% to 95.79% (Figure 7b). Despite the mechanical stress of MACS and FACS, we could show that 75.41% of the resulting purified microglia fraction were viable (Figure 7c). 

Further, in order to prove that the isolated microglia in our protocol were not contaminated by other CD11b^+^ immune cells (e.g., monocytes, macrophages, dendritic cells, natural killer cells, and granulocytes), we performed Ly6C/G staining of the sorted population [49,50,51,52] (Appendix A). Our analysis showed that while neutrophils expressed high levels of both Ly6C and Ly6G, Ly6 expression by the isolated microglia fraction was almost congruent with the FMOs (Table 5, Appendix A). 

For all other CNS-resident cell types, adapting the protocol established in naïve mice to EAE mice led to similar results regarding viability and purity of the isolated single-cell suspensions (Figure 7d–f, Appendix A).

## 4. Discussion

Simultaneously isolating all CNS-resident cell types from one homogenate offers many advantages enabling multi-omic analyses from one single CNS as well as the accurate investigation of complex cellular networks ex vivo. Artifacts caused by using different mice or experimental approaches for the cell isolations are reduced to a minimum, increasing the comparability of cellular analyses. Thus, very importantly, mice numbers can be decreased following the three principles of human experimental techniques replacement, reduction, and refinement, first described by Russell and Burch in 1959 and known as the three Rs in the research community [53].

The advent of the era of systems biology has pushed the limits of cell isolation techniques. Currently, protocols are being developed to map CNS-resident cells ex vivo by combining high-dimensional techniques like RNA sequencing analysis and mass spectrometry [54]. These techniques offer a very accurate cellular profiling in health and disease. However, they are very expensive, require considerable expertise and time-consuming analyses, and do not allow for functional analyses. Microfluidic brain-on-a-chip systems are combining conventional in vitro cultures with an engineered platform allowing the rapid and inexpensive screening for disease mechanisms or testing of new drugs [55,56,57,58]. Yet, similar to 2D and 3D cell cultures, cell growth and migration are geometrically restricted. Also, inter-connections between organs cannot be displayed. Apart from these techniques, it is likely that CNS organoids will gain importance in the near future after their functional output is improved, making it possible to examine inter-cellular connections and interactions as well as cellular modeling under pathological conditions [59,60,61,62,63,64]. 

Overall, until further modifications and simplifications of these new technologies have been implemented and accepted by the broad research society, fluorescence- and magnetic-activated cell sorting represent the most effective methods for the generation of pure and viable single-cell suspensions ex vivo. There are already many publications comparing the advantages of MACS and FACS: Sutermaster et al. showed that MACS leads to increased cell yields and higher cell viability while decreasing the experimental duration [65]. Pan et al. confirmed those findings for microglia and astrocytes [66]. Furthermore, Holt et al. showed that magnetic sorting is a relatively gentle method compared to FACS, preserving cell integrity and retaining a dense network of processes; via MACS, isolated astrocytes showed a complex morphology similar to that pre-sorting and superior when compared to FACS sorted astrocytes [22,67]. Thus, also extrasomatic proteins can be analyzed, representing an advantage of great importance in the CNS where cellular arborizations and processes are known to express different protein signatures than the soma [22,68,69,70,71]. For neurons, Bowles et al. showed that MACS reduces cell stress while improving the yield of viable cells and maintaining a sorting efficiency equivalent to FACS [72]. Also, neuronal populations showed a higher homogeneity after MACS, and their purity in long-term cultures was ameliorated [72]. However, one limitation of the MACS approach is that it requires an antibody against extracellular proteins since there is no established protocol for fixed tissue yet. Apart from that, the complexity of MACS experiments is lower than in FACS approaches, and purchasing the essential equipment is more cost-effective. During the establishment of our protocol, we also observed that FACS experiments significantly decreased cell yields, leading to higher numbers of required mice per experiment. This increased the experimental duration, which in turn compromised cellular viability. Most likely due to the mechanical stress during the FACS sorting and the fact that unlike MACS, it cannot be performed under sterile conditions, consecutive cultivation was much more difficult with many cells detaching and dying after seeding. Therefore, we chose MACS technology for our protocol and combined it with flow cytometry for purity analyses.

In contrast, in the context of neuroinflammation (here: EAE), MACS of microglia had to be followed by further fluorescence-activated cell sorting to eliminate other infiltrated CD11b^+^ immune cells (e.g., monocytes, macrophages, dendritic cells, natural killer cells, and granulocytes). Distinguishing microglia from other CNS-resident myeloid populations by their relatively lower expression of the leukocyte antigen CD45 represents a well-established gating strategy for FACS sorting [49,73,74,75,76,77]. Interestingly, microglia are reported to change their expression levels of the leukocyte antigen CD45 upon activation: Microglia in steady-state are characterized as CD45^low^CD11b^+^ [49,75,76]. However, upon inflammation, their expression of CD45 increases to a CD45^int^ state [49,75,76]. We used these studies as the scientific foundation for our FACS gating strategy for microglia in EAE mice: Firstly, all CNS-resident CD11b^+^ cells were isolated via MACS just as in naïve mice (Figure 1 and Figure 3a). Secondly, the CD45^int^CD11b^+^ population was sorted via FACS (Figure 1 and Figure 7). Since EAE represents a state of inflammation and microglia are very susceptible to activation, most of the CNS-resident microglia were expected to be in an inflamed state. In fact, our flow cytometry analyses of the CD11b^+^ population before further sorting confirmed that only a small minority of the microglia isolated via MACS showed CD45^low^ expression levels (Figure 7b). 

Furthermore, Ly6C and Ly6G staining of the resulting CD45^int^CD11b^+^ population were performed, disproving any contamination by other CD11b^+^ immune cells (Supplemental Figure 1f,g). Lymphocyte antigen 6 (also known as GR1) is a known leukocyte marker with its expression levels depending on the stage of cell differentiation [49,50,51,52]. For example, inflammatory murine monocytes are often defined as CD45^high^CCR2^+^CX_3_CR1^low^GR1^+^, whereas resident monocytes are described as being CD45^high^, CCR2^−^, CX_3_CR1^high^ and GR1^−^ [49,78,79,80,81]. In our study, we used neutrophils as positive controls as they are known to express several Ly6 proteins, including Ly6C and Ly6G (Appendix A) [51]. Gating strategies were based on FMOs. Since Ly6 expression levels of all CD45^int^CD11b^+^ cells were almost congruent with those of the FMOs, our conclusion was that gating of microglia as CD45^int^CD11b^high^ represents a suitable method to avoid significant contamination by other CD11b^+^ immune cells during FACS sorting. Of course, in order to investigate both resting and inflamed microglia, one could additionally sort all CD45^low^CD11b^+^ cells for comparative analyses. Apart from that, there are some new phenotypic and cytokine markers for microglia in the disease state that could be included in future studies (e.g., MHC-II^int^ and CD11c^int^) [49,82,83,84,85,86,87]. 

In general, the search for the best-suited microglial marker has not ended yet and is still being heavily discussed in the scientific field. To date, microglia have been characterized by the expression of Tmem119, Siglec, Slc2a5, P2ry12, Fcrls, SalI1, Hexb, and Trem2 [49,88,89,90]. The transmembrane protein 119 (Tmem119), as well as the P2Y purinoreceptor 12 (P2ry12), represent homeostatic microglia-specific markers suitable to distinguish between microglia, monocytes, and macrophages [49,77,91,92,93]. However, their expression is reported to be downregulated upon activation, reducing their significance for identification of microglia during EAE [49,87]. In contrast, Bennett et al. reported that Tmem119 expression correlates with microglial maturity postnatally and remains stable after injury or inflammation [92]. Interestingly, Masuda et al. very recently proved that the lysosomal enzyme Hexb represents a stably expressed microglia signature gene in several models of neurodegeneration and autoimmune neuroinflammatory conditions [87,94]. We complemented our data with the immunocytochemistry of Iba-1 (Figure 6a). Iba-1 is a well-established pan-microglial marker essential for membrane ruffling and phagocytosis in activated microglia [45,49,86,95]. Its expression increases with microglial activation, e.g., in multiple sclerosis and Alzheimer’s disease [44,86,96,97]. In our study, both CD11b and Iba-1 staining confirmed the purity and cell-type-specific morphology of microglia after isolation (Figure 6a, Appendix A).

Another aspect of this study open to discussion is the fact that mostly female C57BL/6J mice were used. Han et al. stated that male and female mice—irrespective of their strain background—developed comparable EAE courses and cumulative scores [98]. Since our MOG immunization protocol has been optimized for female mice and their group housing is more convenient, we applied the depicted protocol to female C57BL/6J mice. Still, during the establishment of the protocol, we also used naïve male mice with no observable effect on the resulting purities or cell yields. The number of immune cells infiltrating the CNS as well as their cytokine expression is believed to be higher in female EAE mice than in their male counterparts [98], even simplifying the application of our protocol to male mice with the CD11b^+^ population being smaller. Published sex-specific functional differences of microglia represented by significant shifts in their transcriptomic and proteomic profiles [99,100] need to be kept in mind for downstream applications of the protocol but should have no effect on the cell isolation process itself. In fact, this information could be used as an opportunity to apply the here described protocol for a comprehensive multi-omic analysis of male and female microglia in health and disease (e.g., EAE). A recent study already described that female microglia showed a neuroprotective phenotype while male microglia were more susceptible to inflammatory reactions making further analyses of gender-dependent omic shifts even more attractive [100].

In conclusion, the described protocol for the simultaneous isolation of all major CNS-resident cell types from one CNS homogenate offers the following main advantages: A positive influence on the three Rs of animal experiments: Replacement, Reduction, and Refinement.The assessment of cell-cell interactions and characteristics on an individual level. A way to overcome variability within groups: correlating phenotypic characteristics (e.g., behavioral scores) with comprehensive ex vivo analyses of all four major CNS cell types from the same individual animal.The chance to investigate complex cellular networks, e.g., neuronal networks and neuroinflammatory pathways ex vivo.The feasibility of multi-omic analyses from one individual CNS homogenate.The prospect of studying CNS-resident cells through different stages of a disease course, e.g., neuroinflammation, neurodegeneration, and remission in EAE or other disease models.The option of cultivating a fraction of the isolated adult CNS-resident cells in monocultures, allowing for further targeted functional assays.

To the best of our knowledge, our protocol provides significant conceptual advances with far-reaching implications for preclinical and clinical research questions and presents a novel experimental approach for the neuroscientific community.

## Figures and Tables

**Figure 1 cells-10-00651-f001:**
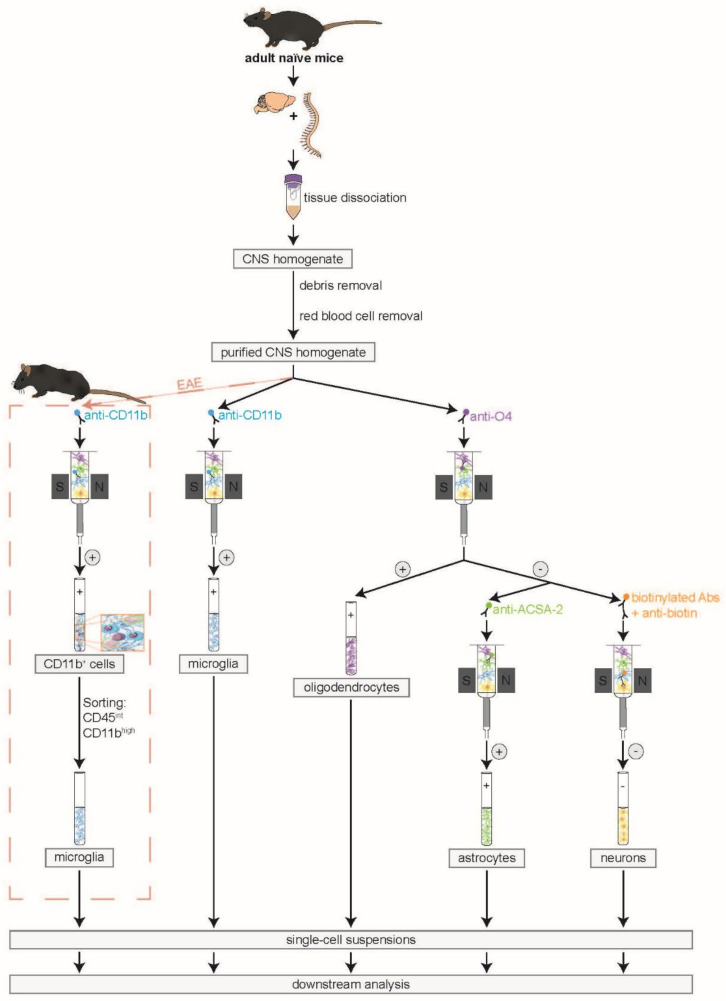
Workflow for simultaneous isolation of all principal CNS-resident cell types. First, the adult brain and spinal cord were dissected and dissociated. The resulting CNS homogenate was further modified by performing debris and red blood cell removal. The purified CNS homogenate was split into two fractions. One fraction was used for MACS of microglia via anti-CD11b MicroBeads (positive selection). Simultaneously, oligodendrocytes were isolated from the second cell fraction using anti-O4 MicroBeads (positive selection). Their negative flow-through was collected and used for the simultaneous isolation of astrocytes with anti-ACSA-2 MicroBeads (positive selection) and neurons by biotin labeling and depletion of all non-neuronal cells (negative selection). In experimental autoimmune encephalomyelitis (EAE) mice, the magnetic labeling of CD11b^+^ cells had to be followed by fluorescence-activated cell sorting of CD45^int^CD11b^high^ cells in order to preserve purity at approximately 90%. Otherwise, CD11b^+^ immune cells like monocytes, macrophages, dendritic cells, natural killer cells, and granulocytes would contaminate the microglia fraction since these cells are known to immigrate into the CNS during the EAE course [41,42,43].

**Figure 2 cells-10-00651-f002:**
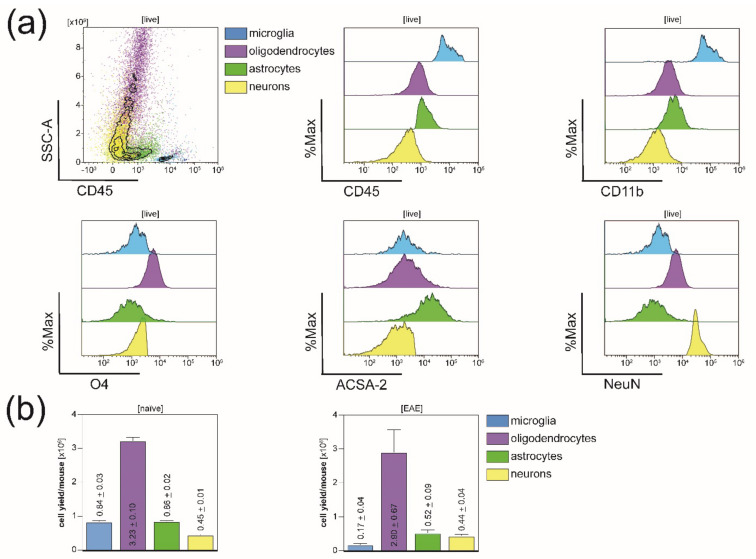
Phenotypic characterization pre-isolation and resulting cell yields post-isolation. (**a**) Flow cytometry analysis of one representative purified CNS homogenate pre-isolation using the same cell-type-specific markers included in panels for purity analyses post-isolation. (**b**) Cell yields per mouse and cell-type after applying the described protocol in naïve (on the left) and in EAE mice (on the right). Five biological replicates were measured for naïve mice, and four biological replicates were acquired in EAE mice. Respective means ± SEMs are indicated within the bar graphs.

**Figure 3 cells-10-00651-f003:**
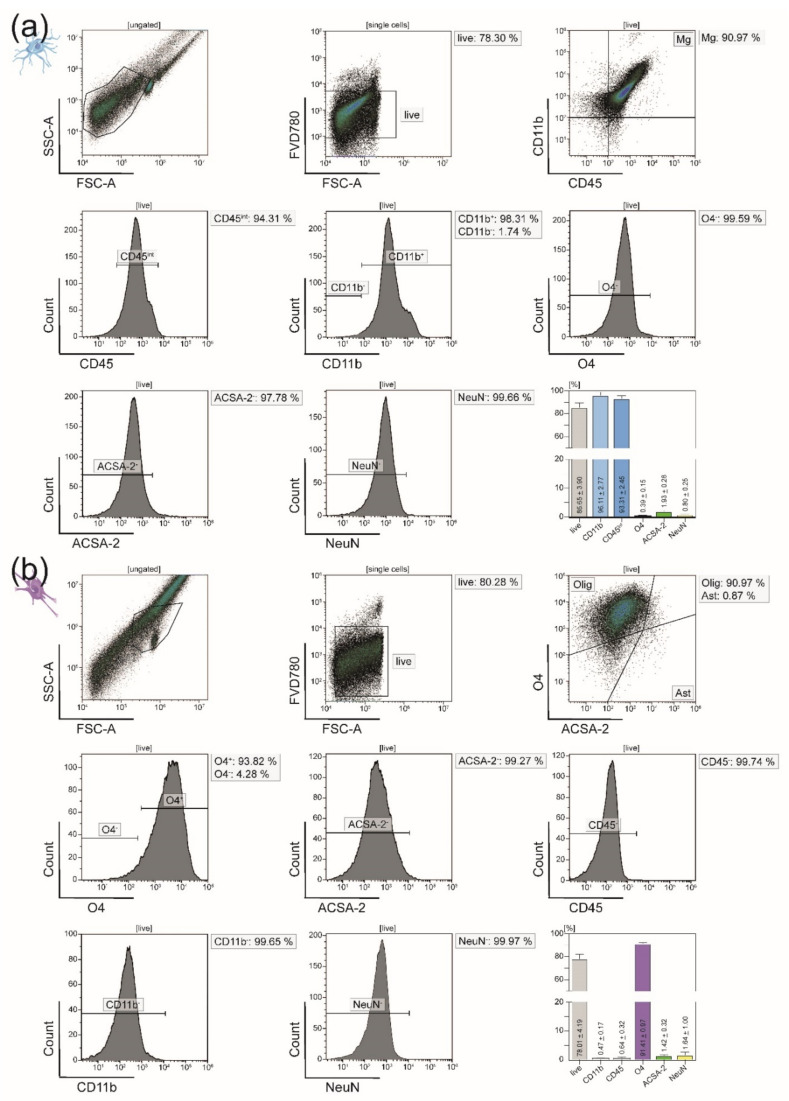
Flow cytometry analyses of microglia and oligodendrocytes isolated from adult naïve mice. (**a**) Exemplary purity analysis of microglia isolated from adult naïve mice via anti-CD11b MicroBeads. Microglia are CD45^int^CD11b^high^ cells and comprised 93.31% of the isolated cell fraction. (**b**) Representative purity analysis of oligodendrocytes isolated from adult naïve mice using anti-O4 MicroBeads. Of all viable isolated cells, 91.41% expressed the oligodendrocyte-specific surface marker O4. Bar graphs visualize the viability and purity of the resulting single-cell suspensions. The depicted cell-type-specific markers were selected for the purity panel of all four major CNS-resident cell types. Five biological replicates were measured for both microglia (**a**) and oligodendrocytes (**b**). Means ± SEMs are indicated within the bars.

**Figure 4 cells-10-00651-f004:**
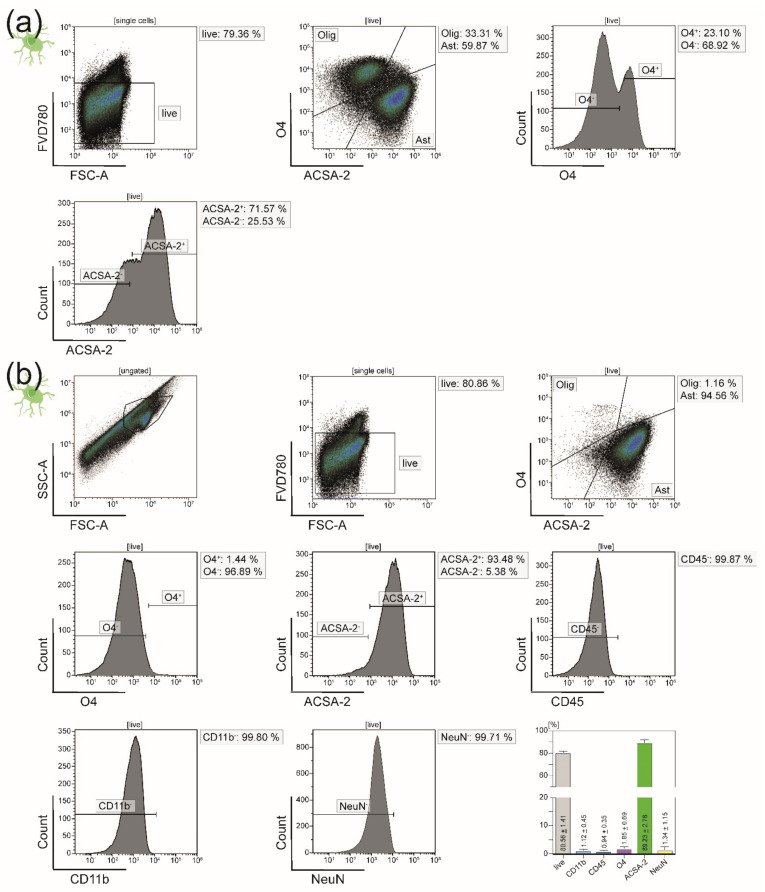
Flow cytometry analyses of astrocytes isolated from adult naïve mice**.** We chose ACSA-2 as a cell-type-specific marker for astrocytes. (**a**) Purity analysis of astrocytes isolated from adult naïve mice via anti-ACSA-2 MicroBeads showing a relevant 33.31% contamination with oligodendrocytes (O4^+^ cells). Only 59.87% of the isolated cells were believed to be astrocytes. (**b**) Representative purity analysis of astrocytes after isolation from the negative flow-through of oligodendrocytes. Oligodendrocytes were depleted successfully, leaving only 1.85% O4^+^ cells behind and thereby increasing the purity of astrocytes to 89.23%. The bar diagram illustrates the percentage of viable and ACSA-2 expressing astrocytes. Five biological replicates were measured. Means ± SEMs are depicted within the bars.

**Figure 5 cells-10-00651-f005:**
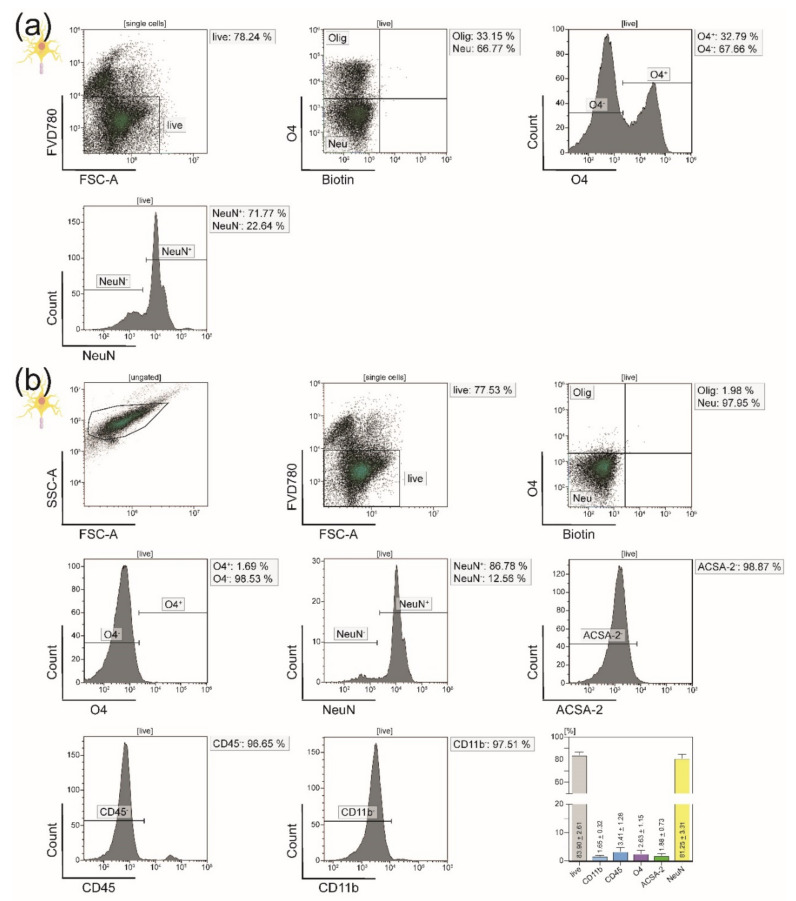
Flow cytometry analyses of neurons isolated from adult naïve mice. We used NeuN as a cell-type-specific marker for neurons. (**a**) Exemplary purity analysis of neurons isolated from adult naïve mice by depletion of all non-neuronal cells after labeling them with biotinylated antibodies and anti-biotin MicroBeads. Of the resulting single-cell suspension, 32.79% expressed the oligodendrocyte-specific surface marker O4. (**b**) Upon isolation of neurons from the negative flow-through of oligodendrocyte, the purity was increased to 81.25%. The bar diagram illustrates the viability and purity of the resulting single-cell suspension. Only 2.63% of the viable isolated cells were still O4^+^ after the protocol improvement. Five biological replicates were measured. Means ± SEMs are indicated in the bar graph.

**Figure 6 cells-10-00651-f006:**
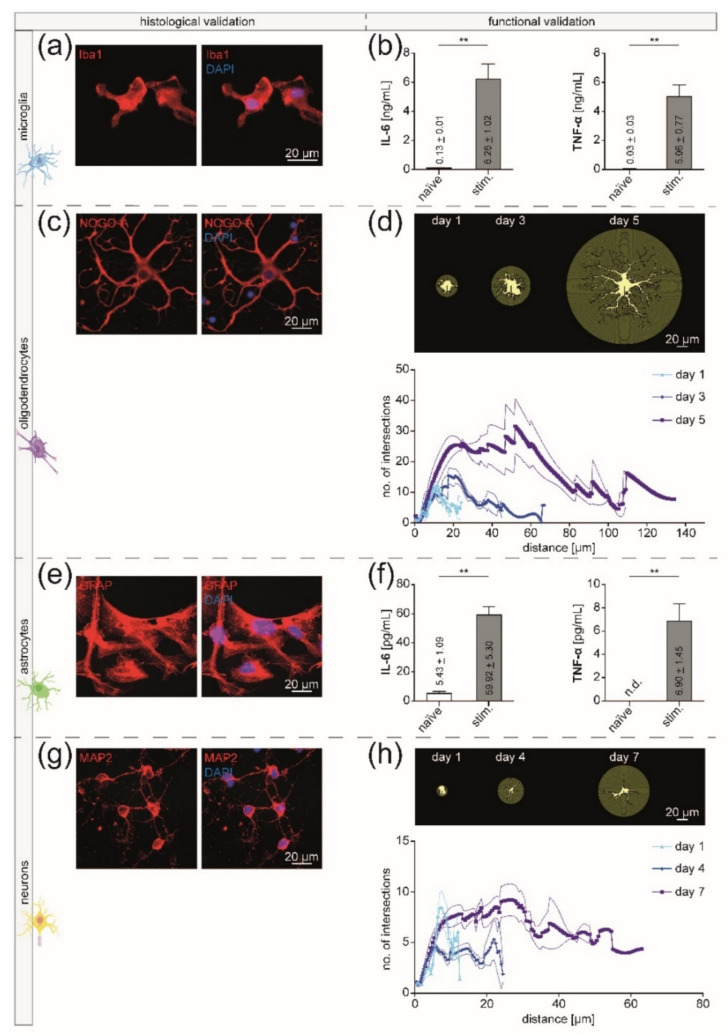
Viability of isolated adult CNS-resident cells after MACS. (**a**,**c**,**e**,**g**) Histological validation of the four isolated CNS-resident cell types using immunocytochemistry. Fluorescence images were acquired with a Zeiss Axio Scope.A1 using 40- (**c**,**e**,**g**) or 63- (**a**) fold objectives and optimal exposure times. (**b**,**d**,**f**,**h**) Functional validation of the four cell types by analyzing the change in their cytokine expression profiles upon stimulation (**b**,**f**) and observing their morphological development during cultivation (**d**,**h**). (**a**) Immunofluorescence staining of microglia with a primary antibody directed against Iba-1. (**b**) The expression levels of the pro-inflammatory cytokines IL-6 and TNF-*α* by microglia were significantly increased after stimulation with 100 ng/mL LPS for 48 h. Per experimental condition, five biological replicates were measured in technical duplicates. For the Mann–Whitney U test, the exact *p*-value was 0.0079. Means ± SEMs are indicated in the bar graphs. (**c**) Oligodendrocytes were stained for their expression of NOGO-A. (**d**) Sholl analysis of oligodendrocytes on days one, three, and five of cultivation. The distance of the concentric shells from the cell center in µm is depicted against the number of their intersections with the cell processes at each radius step size (step size = 0.5 µm). For each time point, five biological replicates were acquired. For each biological replicate, the mean of five cells was calculated displayed as the bold curves. The narrow curves represent the SEM. (**e**) GFAP was used as a cell-type-specific marker for astrocytes. (**f**) IL-6 and TNF-*α* ELISA of astrocytes after incubation with 100 ng/mL LPS for 48 h. Per experimental condition, five biological replicates were measured in technical duplicates. For the Mann–Whitney U test, the exact *p*-value was 0.0079. Means ± SEMs are depicted in the bars. (**g**) MAP2 was used for immunofluorescence staining of neurons. (**h**) Sholl analysis of neurons on days one, four, and seven of cultivation. The distance of the concentric shells from the cell center in µm is depicted against the number of their intersections with the cell processes at each radius step size. For each time point, five biological replicates were acquired. For each biological replicate, the mean of five cells was calculated displayed as the bold curves. The narrow curves represent the SEM.

**Figure 7 cells-10-00651-f007:**
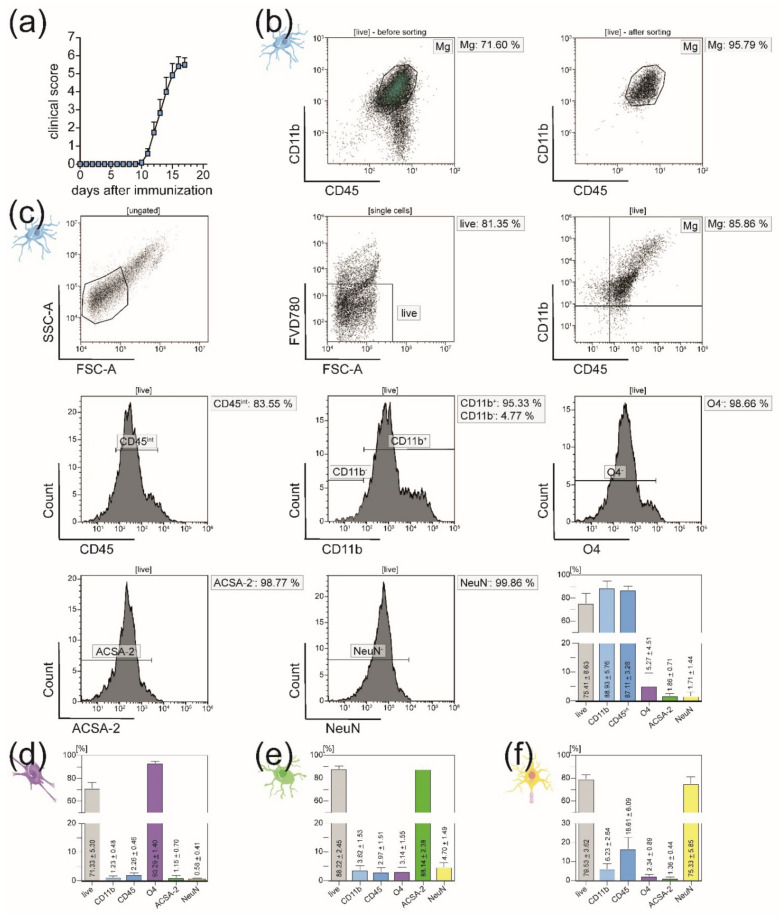
Flow cytometry analyses of CNS-resident cells isolated from adult EAE mice. (**a**) Clinical course of EAE mice (group size = 12 mice). Error bars show mean ± SEM. (**b**) Purity of isolated microglia measured with BD FACSAria III before and after sorting of CD45^int^CD11b^high^ cells from CD11b^+^ single-cell suspension. Via FACS, purity was increased from 71.60% to 95.79%. (**c**) Representative purity analysis of sorted CD45^int^CD11b^high^ cell fraction with flow cytometry purity panel used for all four major CNS-resident cell types. The bar graph illustrates the percentage of viable target cells within the resulting single-cell suspension (*n* = 4). Means ± SEMs are indicated in the bar graphs. (**d**–**f**) Bar diagrams summing up the viability and purity after isolation of oligodendrocytes (**d**; *n* = 6), astrocytes (**e**; *n* = 6), and neurons (**f**; *n* = 5) from adult EAE mice applying the same MACS protocol used for naïve mice**.** Means ± SEMs are depicted in the bar graphs.

**Table 1 cells-10-00651-t001:** Protocol for simultaneous MACS of microglia and oligodendrocytes from naïve and EAE mice. Steps that should be performed at once are listed in the same row.

	Microglia	Oligodendrocytes
	Positive Selection	Positive Selection
1. Blocking	Starting volume was adjusted to 90 µL PB buffer per 1 × 10^7^ cells. Cells were resuspended carefully.	Starting volume was adjusted to 80 µL PB buffer per 1 × 10^7^ cells. Cells were resuspended carefully.
	Incubation for 10 min on ice.	FcR Blocking Reagent (mouse; Miltenyi Biotec) was added, 10 µL per 1 × 10^7^ cells. Incubation for 10 min at 2–8 °C.
2. Magnetic labeling	CD11b MicroBeads (human, mouse; Miltenyi Biotec) were applied, 10 µL per 1 × 10^7^ cells, and mixed with the cell suspension.	Anti-O4 MicroBeads (human, mouse, rat; Miltenyi Biotec) were applied, 10 µL per 1 × 10^7^ cells, and mixed with the cell suspension.
	Incubation for 15 min in the dark at 2–8 °C.	
3. Washing and preparation of columns	Cells were washed by adding 2 mL PB buffer per 1 × 10^7^ cells followed by centrifugation at 300 g and 4 °C for 10 min.	
	Meanwhile, one LS Column per 4 × 10^7^ cells was placed with its column wings to the front in one of four gaps of a QuadroMACS Separator attached to a MultiStand (Miltenyi Biotec). Each column was equilibrated with 3 mL PB buffer. The flow-through was discarded.	
6. Magnetic separation—negative fraction	Upon termination of the centrifugation, the supernatant was aspirated completely. Up to 1 × 10^7^ cells were resuspended in 500 µL PB buffer. The cell suspension was distributed among the utilized number of columns per cell type. The flow-through containing the unlabeled cells was collected in a new 15 mL falcon placed on ice underneath each column.	
	The columns were washed three times with 3 mL PB buffer. The flow-through was pooled together with the unlabeled cells from the previous step in the 15 mL falcon. This negative fraction contained all non-targeted cells (CD11b^−^, O4^−^ cells).	
7. Magnetic separation—positive fraction	The columns were removed from the magnetic field and placed on a new 15 mL falcon. PB buffer was added to each LS column (5 mL). The magnetically labeled cells were immediately flushed out by firmly pushing the provided plunger into the column.	
	This positive fraction contained all targeted cells (microglia or CD11b^+^ cells and oligodendrocytes or O4^+^ cells).	
8. Counting of targeted cells	The total number of isolated cells was determined with the help of the electronic cell counter and analyzer system CASY TT (Roche Innovatis AG, Bielefeld, Germany).	
	Settings for counting of microglia: Capillary: 150 µm, sample vol: 400 µL, x-axis: 30 µm, cycles: 3, dilution 2.01 × 10^2^, y-axis: auto. Eval. Cursor: 8.03–30.00 µm, norm. cursor: 4.88–30.00 µm. % Calculation: % via, debris: off.	
	Settings for counting of oligodendrocytes: Capillary: 150 µm, sample vol: 400 µL, x-axis: 30 µm, cycles: 3, dilution 5.01 × 10^2^, y-axis: auto. Eval. Cursor: 8.03–30.00 µm, norm. cursor: 6.00–30.00 µm. % Calculation: % via, debris: off.	

**Table 2 cells-10-00651-t002:** Protocol for simultaneous MACS of neurons and astrocytes from naïve and EAE mice. Steps that should be performed at once are listed in the same row.

	Neurons	Astrocytes
	Negative Selection	Positive Selection
1. Blocking	Starting volume was adjusted to 80 µL PB buffer per 1 × 10^7^ cells. Cells were carefully resuspended.	Starting volume was adjusted to 80 µL AstroMACS Separation buffer (Miltenyi Biotec) per 1 × 10^7^ cells. Cells were carefully resuspended.
	Incubation for 10 min on ice.	FcR Blocking Reagent was added, 10 µL per 1 × 10^7^ cells. Incubation for 10 min at 2–8 °C.
2. Biotinylation of non-neuronal cells, magnetic labeling of astrocytes	Non-Neuronal Cell Biotin-Antibody Cocktail (mouse; Miltenyi Biotec) was applied, 20 µL per 1 × 10^7^ cells, and mixed with the cell suspension.	Anti-ACSA-2 MicroBeads (mouse; Miltenyi Biotec) were applied, 10 µL per 1 × 10^7^ cells, and mixed with the cell suspension.
	Incubation for 5 min in the dark at 2–8 °C.	Incubation for 15 min in the dark at 2–8 °C.
	PB buffer was added, 1 mL per 1 × 10^7^ cells, followed by centrifugation at 300× *g* and 4 °C for 5 Min. The supernatant was aspirated completely. The cell pellet was resuspended in 80 µL PB buffer per 1 × 10^7^ cells.	
3. Magnetic labeling of biotinylated cells, washing of astrocytes, and preparation of columns	Anti-Biotin MicroBeads (mouse; Miltenyi Biotec) were added, 20 µL per 1 × 10^7^ cells, and mixed with the cell suspension.	Cells were washed by adding 1 mL AstroMACS Separation buffer per 1 × 10^7^ cells followed by centrifugation at 300× *g* and 4 °C for 10 min.
	Incubation for 10 min in the dark at 2–8 °C.	
	Meanwhile, one LS Column per 4 × 10^7^ cells was placed with its column wings to the front in one of four gaps of a QuadroMACS Separator attached to a MultiStand.	
	Each column was equilibrated with 3 mL PB buffer. The flow-through was discarded.	Each column was equilibrated with 3 mL AstroMACS Separation buffer. The flow-through was discarded.
6. Magnetic separation—negative cell fraction	Upon termination of the centrifugation, the supernatant was aspirated completely.	
	Up to 1 × 10^7^ cells were resuspended in 500 µL PB buffer.	Up to 1 × 10^7^ cells were resuspended in 500 µL AstroMACS Separation buffer.
	The cell suspension was distributed among the utilized number of columns per cell type. The flow-through containing the unlabeled cells was collected in a new 15 mL falcon placed on ice underneath each column.	
	The columns were washed twice with 1 mL PB buffer.	The columns were washed three times with 3 mL AstroMACS Separation buffer.
	The flow-through was pooled together with the unlabeled cells from the previous step in the 15 mL falcon.	
	This negative fraction contained all targeted cells (neurons or Biotin^−^ cells).	This negative fraction contained all non-targeted cells (ACSA-2^−^ cells).
7. Magnetic separation—positive cell fraction	The columns were removed from the magnetic field and placed on a new 15 mL falcon.	
	PB buffer was added to each LS column (5 mL).	AstroMACS Separation buffer was added to each LS column (5 mL).
	The magnetically labeled cells were immediately flushed out by firmly pushing the plunger into the column.	
	This positive fraction contained all non-targeted cells (non-neuronal or Biotin^+^ cells).	This positive fraction contained all targeted cells (astrocytes or ACSA-2^+^ cells).
8. Counting of targeted cells	The number of isolated cells was determined with the help of the electronic cell counter and analyzer system CASY TT (Roche Innovatis AG, Bielefeld, Germany).	
	Settings for counting of neurons: Capillary: 150 µm, sample vol: 400 µL, x-axis: 30 µm, cycles: 5, dilution 1.00 × 103, y-axis: auto. Eval. Cursor: 8.03–25.80 µm, norm. cursor: 8.03–30.00 µm. % Calculation: % via, debris: off.	
	Settings for counting of astrocytes: Capillary: 150 µm, sample vol: 400 µL, x-axis: 20 µm, cycles: 3, dilution 2.01 × 10^2^, y-axis: auto. Eval. Cursor: 6.95–20.00 µm, norm. cursor: 4.90–20.00 µm. % Calculation: % via, debris: off.	

**Table 3 cells-10-00651-t003:** Cultivation of isolated CNS-resident cells.

Header	Microglia	Astrocytes	Oligodendrocytes	Neurons
Coating	Four- or 24-wells were coated with 0.5% poly-L-lysine hydrobromide (Merck KGaA) overnight at 4 °C.			Four- or 24-wells were coated with 50 µg/mL poly-D-lysine hydrobromide (Merck KGaA) overnight at 4 °C.
	Dishes were washed once with ddH_2_O before plating of cells.	Dishes were washed three times with ddH_2_O followed by a second coating with 10 µg/mL laminin for two hours at 37 °C. Subsequently, dishes were washed once with ddH_2_O before plating of cells.	Dishes were washed once with ddH_2_O before plating of cells.	
Medium	Dulbecco’s Modified Eagle’s Medium (Thermo Fisher Scientific)		MACS Neuro Medium (Miltenyi Biotec)	
	+10% fetal bovine serum		+2% MACS NeuroBrew-21 (Miltenyi Biotec)	
	+1% non-essential amino acids solution (Thermo Fisher Scientific)		+1% penicillin-streptomycin	
	+1% penicillin-streptomycin		+0.25% GlutaMAX supplement	
	+0.1% 2-mercaptoethanol			
	+1% GlutaMAX supplement (Thermo Fisher Scientific)			
		+0.2% AstroMACS supplement (Miltenyi Biotec)	+10 ng/mL human platelet derived growth factor AA (PeproTech, NJ, USA)	
			+10 ng/mL human fibroblast growth factor 2 (PeproTech)	
			+5 ng/mL human neurotrophin-3 (PeproTech)	
Cultivation	Per well, 2 × 10^5^ cells were seeded.	Per well, 3 × 10^5^ cells were seeded.		
	The isolated cells were resuspended in warm medium and plated as a drop of 50 µL in the middle of each coated 24-well. After incubation for 45 min in a 37 °C incubator (5% CO_2_), allowing the cells to settle down, 450 µL of warm medium was added carefully to each well.			
	On the next morning, the medium was changed to remove all cell debris caused by the MACS procedure.			The whole medium was replaced immediately to remove all non-attached and dead cells.
	Continuous cultivation at 37 °C (5% CO_2_). No further medium changes were performed afterward. Cells were inspected daily to check for their confluence and morphology.			
Further processing	Microglia were processed on day 2 of cultivation.	The cultivated cells were processed on day 5 of cultivation.		Neurons were processed on day 7 of cultivation.

**Table 4 cells-10-00651-t004:** Antibodies used for fluorescence stainings.

Cell Type	Primary Antibody (Host, Dilution; Company)	Secondary Antibody (Host, Dilution; Company)
Microglia	Ionized calcium-binding adapter molecule 1 (Iba-1, rabbit, 1:2000; FUJIFILM Wako Chemicals GmbH, Neuss, Germany)	AF594 (goat, 1:500; Thermo Fisher Scientific)
	CD11b (rat, 1:100; Bio-Rad Laboratories, Santa Rosa, CA, USA)	Cy3 (donkey, 1:500; Dianova, Hamburg, Germany)
Oligodendrocytes	NOGO-A (rabbit, 1:200; Merck KGaA)	AF594 (goat, 1:500; Thermo Fisher Scientific)
Astrocytes	Glial fibrillary acidic protein (GFAP, rabbit, 1:1000; Abcam)	Cy3 (goat, 1:300; Jackson ImmunoResearch Laboratories, West Grove, PA, USA)
Neurons	Microtubule-associated protein 2 (MAP2, rabbit, 1:200; Santa Cruz Biotechnology, Street Dallas, TX, USA)	Cy3 (donkey, 1:500; Dianova)

**Table 5 cells-10-00651-t005:** Mean fluorescence intensities (MFIs) of Ly6C/G staining.

	FMO	Microglia	CD45^high^CD11b^+^
Ly6C	1,499.65	1,733.91	5,735.73
Ly6G	1,497.07	1,803.32	5,485.70

## Data Availability

Not applicable.

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
