# Peer review of "One Brain—All Cells: A Comprehensive Protocol to Isolate All Principal CNS-Resident Cell Types from Brain and Spinal Cord of Adult Healthy and EAE Mice"

_cells, 2021, doi:10.3390/cells10030651_

Round 1
Reviewer 1 Report
In this study (cells-1067670), Schroeter and colleagues modified the protocols in order to simultaneously isolate microglia, oligodendrocytes, astrocytes and neurons from the same CNS homogenate. Some concerns and suggestions are listed below:
- How many cell numbers of microglia, oligodendrocytes, astrocytes and neurons can you obtain from one brain (or half) by using this method?
- Did you compare the effectives between this modified method (MACS sorting) and FACS sorting?
- In this study, only female wild type C57BL/6J mice were used. However, recent data obtained from preclinical models indicated that the number and phenotype of microglia may differ between females and males (Guneykaya et al. Transcriptional and translational differences of microglia from male and female brains. Cell Rep. 2018, 24, 2773–2783; Villa et al. Sex-specific features of microglia from adult mice. Cell Rep. 2018, 23, 3501–3511). Furthermore, an underestimated yet marked sex-dependent microglial activation pattern may exist in the injured CNS during EAE (Han et al, Sex-specific effects of microglia-like cell engraftment during experimental autoimmune encephalomyelitis, International Journal of Molecular Sciences, 2020 Sep 17;21(18):6824). This point should be discussed.
- The authors said that microglia were gated as CD45intCD11bhigh. However, additional markers such as Ly6C-Ly6G- should also be used for identifying microglia (Figure 6).
- It is generally thought that the level of CD45 on the CD11b positive population is an effective way to distinguish microglia from macrophages. However, this method relies on relative surface marker expression as assessed by flow cytometry and has the serious limitation that CD45 expression can change in the context of inflammation (such as EAE).
- In Figures 2-4, some pictures lack outer contour lines. This may be caused by using the Adobe Illustrator. Please revised the figures accordingly.
- In healthy condition, how perivascular, subdural meningeal and choroid plexus macrophages in the CNS can be isolated?
- In Figure 5a, CD11b is not a good marker for microglia. The authors also said that other immune cells may also express CD11b. Therefore, P2ry12 or Tmem119 should be used.
- The part of discussion is too short. More information should be added and discussed.
- In Figure 5, qPCR or Western blot should be performed.
- A similar protocol has already been published (Holt et al, Magnetic cell sorting for in vivo and in vitro astrocyte, neuron, and microglia analysis, Curr Protoc Neurosci).
Reviewer 2 Report
This manuscript describes a method to isolate four non-immune CNS-resident cell types (i.e. astrocytes, microglia, neurons and oligodendrocytes) in naïve and EAE mice. It is well written and has its merits, especially as it shows a way to isolate these cell types “simultaneously” on the same CNS homogenate of one single mouse, using the MACS technology instead of FACS (thus allowing to recover less “stressed” cells). However, I believe one crucial parameter is missing and should be included in the analysis: cell yield. Indeed, the fact that isolation can be done from one mouse is of interest, however the manuscript could be improved if authors would show what is the absolute number of the specified isolated cells after completion of the protocol – this would be of great interest and useful to the readers.
Besides the aforementioned remark, I believe it would be good to show phenotypic characterization pre- and post- isolation. Further, it would be a good rationale for the gating strategy used for marker positivity, which lacks negative control (e.g. control isotype antibodies and/or negative cells) for most of the histograms/dot plots and sometimes looks a bit random/arbitrary.
I believe the experiments were repeated several times: number of biological replicates should be mentioned, and mean ±SD/SEM should appear on bar graphs (representing viability and purity) in order to give an idea of the reproducibility of the technique.
Minor point: even if not crucial, I believe it could be good to show what is the composition of the non-neuronal cell biotin-antibody cocktail during negative selection of neurons.
Round 2
Reviewer 1 Report
The authors have addressed my conerns.
Regarding the question 1, why the cell yield per mouse was decreased in EAE (170.000 microglia VS 840.000 microglia)?
Reviewer 2 Report
I believe the authors made good improvements in this revised version of the manuscript. In particular, adding the absolute yield is a major improvement.
Here are my comments:
The explanation for the gating strategy based on FMO does not look sufficient to me. Indeed, the authors provided us FMO staining, but we do not know on which kind of cells it was obtained. FMO could greatly vary depending on the cell type analyzed. Actually this is even more obvious since the gating strategy changed depending on the cell type analyzed (eg microglia are considered O4+ starting from 104 fluorescence, while for oligodendrocytes it is from 2.102 ). If authors based their strategies on FMO for each cell type, then they could easily show on the same graph an overlap of the FMO histogram (empty histogram with dashed lines) with the one of the complete staining (solid lines and histograms): this would justify the difference in gating strategy observed for different cell types that may look suspicious to readers.
I have some issues at understanding what the fsc/ssc gating means on Fig3 and 4 – and whether it was used for the rest of the analyses (it should not, otherwise it would represent a bias in purity analysis).
